# Frontal Transcranial Direct Current Stimulation in Moderate to Severe Depression: Clinical and Neurophysiological Findings from a Pilot Study

**DOI:** 10.3390/brainsci15060540

**Published:** 2025-05-22

**Authors:** Florin Zamfirache, Gabriela Prundaru, Cristina Dumitru, Beatrice Mihaela Radu

**Affiliations:** 1Department of Anatomy, Animal Physiology, and Biophysics, Faculty of Biology, University of Bucharest, Splaiul Independentei, No. 91–95, 050095 Bucharest, Romania; zamfirache.florin@s.bio.unibuc.ro (F.Z.); prundaru.gabriela-narcisa@s.bio.unibuc.ro (G.P.); beatrice.radu@bio.unibuc.ro (B.M.R.); 2Department of Educational Sciences, Faculty of Educational Sciences, Social Sciences and Psychology, Pitesti University Center, The National University of Science and Technology POLITEHNICA Bucharest, Targul din Vale, Nr.1, 110040 Pitesti, Romania

**Keywords:** depression, tDCS, neurophysiological effects, theta/alpha ratio

## Abstract

**Background/Objectives**: Transcranial Direct Current Stimulation (tDCS) has proven to be a promising intervention for major depressive disorder (MDD). Even so, the specific neurophysiological mechanisms underlying its therapeutic effects, particularly regarding frontal EEG markers, remain insufficiently understood. This pilot study investigated both the clinical efficacy and neurophysiological impact of frontal tDCS in individuals with mild to severe depression, with particular focus on mood changes and alterations in Frontal Alpha Asymmetry (FAA), Beta Symmetry, and Theta/Alpha Ratios at the F3 and F4 electrode sites. **Methods**: A total of thirty–one participants were enrolled and completed a standardized Flow Neuroscience tDCS protocol targeting the dorsolateral prefrontal cortex using a bilateral F3/F4 montage. The intervention included an active phase of five stimulations per week for three weeks, followed by a Strengthening Phase with two stimulations per week. Clinical outcomes were assessed using the Montgomery–Åsberg Depression Rating Scale (MADRS), while neurophysiological changes were evaluated via standardized quantitative EEG (QEEG) recordings obtained before and after the treatment course. Among the participants, fourteen individuals had a baseline MADRS score of ≥20, indicating moderate to severe depressive symptoms. **Results**: Following tDCS treatment, significant reductions in MADRS scores were observed across the cohort, with clinical response rates notably higher in the moderate/severe group (71.4%) compared to the mild depression group (20.0%). Neurophysiological effects were modest: no significant changes were detected in FAA or Beta Symmetry measures. However, a substantial reduction in the Theta/Alpha Ratio at F4 was found in participants with moderate to severe depression (*p* = 0.018, Cohen’s d = −0.72), suggesting enhanced frontal cortical activation associated with clinical improvement. **Conclusions**: These findings indicate that frontal tDCS is effective in reducing depressive symptoms, particularly in cases of moderate to severe depression. While improvements in FAA and Beta Symmetry were not significant, changes in the Theta/Alpha Ratio at F4 point toward dynamic neurophysiological reorganization potentially linked to therapeutic outcomes. The Theta/Alpha Ratio may serve as a promising biomarker for tracking tDCS response, whereas other EEG metrics might represent more stable trait characteristics. Future research should prioritize individualized stimulation protocols and incorporate more sensitive neurophysiological assessments, including functional connectivity analyses and task-evoked EEG paradigms, to understand the mechanisms underlying clinical improvements.

## 1. Introduction

Major depressive disorder (MDD) remains one of the leading causes of disability worldwide, affecting over 280 million people and contributing substantially to the global burden of disease [1]. Even with the growing availability of pharmacological and psychotherapeutic interventions, a significant group of individuals with moderate to severe depression either fail to achieve remission or experience undesirable side effects from standard treatments [2,3]. From this basis, there is a growing need for alternative or adjunctive therapies that are both effective and well-tolerated.

Transcranial direct current stimulation (tDCS) has emerged over the past two decades as a promising non-invasive neuromodulatory technique for treating mood disorders. By delivering low-intensity electrical currents to specific cortical areas, tDCS modulates neuronal excitability and functional connectivity within mood-regulating networks [4,5]. Several studies have demonstrated that anodal stimulation over the dorsolateral prefrontal cortex (DLPFC) can significantly improve depressive symptoms, mainly when applied repeatedly over multiple sessions [5,6].

While clinical outcomes have been encouraging, the underlying neurophysiological mechanisms associated with tDCS-induced antidepressant effects remain incompletely understood. One promising avenue for investigation is Frontal Alpha Asymmetry (FAA), measured via quantitative electroencephalography (QEEG). FAA is among the most consistently studied EEG markers in depression. Typically, individuals with depression exhibit greater alpha activity (which reflects lower cortical activation) over the left frontal cortex compared to the right, suggesting left frontal hypoactivation. Treatments that successfully improve depression, including pharmacotherapy, psychotherapy, and neuromodulation techniques such as tDCS, are often associated with normalization or reduction of this asymmetry [7,8].

Meta-analyses have established strong links between left frontal hypoactivation, indexed by FAA, and depressive symptomatology [7]. Furthermore, the FAA has increasingly been conceptualized not only as a static trait marker but also as a dynamic moderator of emotional responses and treatment outcomes [9]. Recent reviews highlight the potential of non-invasive brain stimulation methods such as tDCS and transcranial magnetic stimulation (TMS) in modifying neurophysiological patterns associated with depression, with FAA changes proposed as a key biomarker for monitoring therapeutic effects [10]. These findings support the relevance of EEG markers like FAA for future personalized neuromodulatory interventions. Despite its theoretical and clinical potential, relatively few studies have systematically examined FAA modulation in response to tDCS, particularly in individuals with moderate to severe depression.

The present pilot study aims to explore both clinical efficacy and the neurophysiological correlates of frontal tDCS in patients with moderate to severe MDD. Clinical outcomes are assessed using the Montgomery–Åsberg Depression Rating Scale (MADRS), a widely used measure of depressive symptom severity. At the same time, neurophysiological changes are evaluated by analyzing FAA patterns pre- and post-stimulation. By changing the paradigm to a combined approach between standardized clinical assessments, like MADRS, with QEEG-based neurophysiological measures, this study seeks to provide some preliminary data on the relationship between symptom improvement and modulation of frontal cortical activity, contributing to the effort of a better understanding of the therapeutic mechanisms of tDCS in depression but also in a better way to objectively identify improvements of treatments. Specifically, the objectives of this study are: (1) To evaluate the clinical efficacy of frontal tDCS in individuals with moderate to severe depression compared to those with mild depression, and (2) To investigate neurophysiological changes, particularly alterations in Frontal Alpha Asymmetry (FAA), Beta symmetry and theta/alpha ratios, associated with clinical response following tDCS.

## 2. Materials and Methods

### 2.1. Ethics Statement

Experimental procedures were performed following the University of Bucharest (Romania). They were approved by the University of Bucharest Ethical Committee (Approval No: 14/28 April 2021).

### 2.2. Participants

A total of 31 individuals, aged 18–52, participated in the study: 18 females and 13 males. Fourteen had an initial depression baseline MADRS score ≥ 20, relevant for moderate to severe depression (see Appendix A).

Participants were recruited via an open invitation through social media channels. They were provided with a written description of the research and signed informed consent to participate in the study.

An initial assessment was done for exclusion criteria: ≤18 years, having any implant devices, history of epilepsy or seizures, injuries or defects of the skull, any diagnosed psychosis, history of migraine, or pregnancy.

### 2.3. Transcranial Direct Current Stimulation (tDCS) Protocol and Device Specifications

Transcranial direct current stimulation (tDCS) was administered following the standard protocol developed by Flow Neuroscience, targeting the F3 and F4 sites corresponding to the dorsolateral prefrontal cortex (DLPFC). The intervention consisted of an Active Phase, involving five stimulation sessions per week for three weeks, followed by a Strengthening Phase with two sessions per week. Stimulation was delivered using a headset designed with two prepositioned conductive rubber electrodes (each with a surface area of 23 cm^2^) placed over the forehead. A current of 2 mA was applied for 30 min per session. Impedance was monitored in real time before and throughout each tDCS session using the device’s built-in impedance check feature. Stimulation commenced only when impedance levels were below the safety thresholds specified by the manufacturer. Continuous monitoring ensured stable electrode contact during the session, and electrodes were readjusted if impedance levels rose during stimulation.

The device is a Class II medical device, specifically developed for the treatment of major depressive disorder. To date, it has been utilized by over 15,000 patients across more than 70 healthcare institutions [11]. Electrode placement followed the conventional montage, with the anode positioned over F3 (left DLPFC) and the cathode over F4 (right DLPFC). Device operation, including stimulation parameters, was managed via Bluetooth through the Flow mobile application, which also provided complementary behavioral psychotherapy-based training modules.

### 2.4. Depression Assessment Using the Montgomery–Åsberg Depression Rating Scale (MADRS)

Depressive symptoms were monitored using the self-reported Montgomery–Åsberg Depression Rating Scale (MADRS), with assessments conducted at baseline and after the intervention.

### 2.5. Patient Health Questionnaire (PHQ––9)

Participants were monitored using the Patient Health Questionnaire via the Flow Mobile app. The PHQ-9 questionnaire was reported as a reliable and valid self-evaluation tool for measuring depression disorder severity criteria based on nine symptoms: mood, feeling of unease, sleep, appetite, concentration, initiative, emotional involvement, pessimism, and zest for life [12].

### 2.6. Quantitative Electroencephalography (QEEG) Assessment

A surface quantitative electroencephalography (sQEEG) was performed to generate a brain map and visualize key mental state indicators, including Frontal Alpha Asymmetry (FAA), Beta Symmetry, and Theta/Alpha Ratio. These neurophysiological parameters were measured at baseline and post-treatment using the Muse 2 headband in conjunction with the Myndlift application. The aim of our study was to develop an accessible, user-friendly tool for practitioners, with a specific focus on differences in activity between the left and right frontal hemispheres—given that individuals with depression typically exhibit relatively reduced left compared to right resting frontal activity (Stewart et al., 2012) [13]. Despite its limited number of electrodes, the Muse device has demonstrated the ability to effectively distinguish between high and low valence/arousal emotional states, with an accuracy comparable to that achieved using the full set of DEAP electrodes. The brain map generated represents a visualization of the EEG-specific information from different regions of the brain. These are used to show which areas of the brain are activated more or less compared to a standard, indicating with different colors how far they are from the norm (Figure 1).

### 2.7. Statistical Analysis

Statistical analyses will include paired *t*–tests to evaluate changes in MADRS scores, Frontal Alpha Asymmetry (FAA) in the Beta band (BA), and the Theta/Alpha Ratio (TAC) from baseline to post-treatment. Effect sizes were estimated using Cohen’s d. Pearson correlation analyses were performed to evaluate the relationships between the number of active stimulation weeks, the total number of sessions, and the degree of improvement in MADRS scores. Clinical response threshold was defined as a reduction of 50% or more in MADRS scores from baseline. Defining treatment response as a 50% reduction in MADRS scores aligns with FDA and EMA guidelines and is a widely accepted standard in both pharmacological and neuromodulation research, including studies on rTMS, ECT, and tDCS.

## 3. Results

### 3.1. MADRS and PHQ–9 Score Improvements After tDCS

Participants were divided into two groups by their baseline MADRS scores, using a threshold of ≥20 to indicate moderate to severe depression (Table 1). Among participants with a MADRS score ≥ 20, the results were highly statistically significant, with approximately 71% achieving a clinical response. In addition to age and baseline depression severity, clinical background information was collected. Eighteen participants (58%) reported a history of treatment for depression, and ten (32%) were on stable pharmacological therapy during the study—primarily selective serotonin reuptake inhibitors. Medication regimens were maintained without changes throughout the intervention period. Comorbid psychiatric conditions included anxiety disorders (*n* = 6), mild attention-deficit/hyperactivity disorder (*n* = 2), and somatic symptom presentations (*n* = 3). These factors were not used as exclusion criteria in order to maintain ecological validity, but they were documented and taken into account during data interpretation.

Participants were enrolled in an open-label, real-world pilot study investigating the effects of a standardized tDCS protocol delivered via a commercially available medical device (Flow Neuroscience). Accordingly, no pre-screening based on MADRS score thresholds was conducted at intake, reflecting naturalistic usage patterns in early-intervention contexts where individuals may seek neuromodulation for subthreshold symptoms or relapse prevention. To reduce the potential confounding effect of very low MADRS scores, we structured our analyses to distinguish between participants with moderate to severe depression (MADRS ≥ 20) and those with mild symptoms (MADRS 7–19). Two participants with MADRS scores < 6 were excluded from the clinical response rate analysis of the mild group, and their data were not included in outcome comparisons across depressive severity groups.

When comparing male and female participants, both groups showed significant improvements in MADRS scores following tDCS treatment (Figure 2) (see Appendix A). However, the effect size was smaller among females (*p* = 0.043), while males exhibited stronger clinical effects, with a higher effect size (Cohen’s d = 2.36 for males versus 1.10 for females). The clinical response rate was slightly higher in males (75%) compared to females (67%).

A subgroup analysis was performed to assess sex-specific differences in clinical outcomes following tDCS treatment (Table 2). Among male participants (*n* = 8), MADRS scores showed a significant reduction (t (7) = 6.68, *p* = 0.00028, Cohen’s d = 2.36), with a clinical response rate (≥50% reduction) of 75.0%. Female participants (*n* = 6) also experienced significant improvements (t (5) = 2.70, *p* = 0.043, Cohen’s d = 1.10), with a clinical response rate of 66.7%. Although both groups demonstrated substantial clinical gains, males exhibited slightly greater effect sizes and higher response rates compared to females. While both male and female participants showed significant improvements following tDCS, the effect size was notably greater in males. This may point to sex-specific differences in neuromodulatory responsiveness or in the neural circuits involved in emotion regulation targeted by DLPFC stimulation. However, these findings are preliminary and should be interpreted with caution. Further research with larger sample sizes is needed to better understand the potential moderating role of sex in tDCS treatment outcomes.

Among the remaining participants, two had baseline MADRS scores below 6, reflecting no depressive symptoms, while fifteen had initial scores between 7 and 19, consistent with mild depression (Table 3). Statistical significance remained very strong (*p* < 0.001). The effect size increased slightly, with Cohen’s d rising to 1.40 following the exclusion of participants D.C. and M.D. The clinical response rate decreased slightly to 20%.

Both groups showed significant symptom improvement, with the moderate/severe group exhibiting greater reductions and higher clinical response rates than the mild group (Figure 3). In the mild group, more sessions and longer active treatment phases were associated with greater MADRS improvements, although correlations did not reach significance (*p* ≈ 0.06–0.10). No significant association between treatment duration and symptom change was observed in the moderate/severe group.

As illustrated in Figure 3, tDCS treatment led to statistically significant improvements in mood (*p* = 0.0224, Cohen’s d = 0.87), sleep (*p* = 0.0082, Cohen’s d = 1.07), and concentration (*p* = 0.0059, Cohen’s d = 1.13). These outcomes are supported by large effect sizes, with all Cohen’s d values exceeding 0.8. No significant changes were observed in feelings of unease (*p* = 0.1369) or appetite (*p* = 0.6164). These results further underline the potential of tDCS to enhance key affective and motivational domains, especially in areas such as initiative, emotional involvement, and pessimism, where large effect sizes were also observed.

Pearson correlations were computed to examine the relationship between improvements in MADRS scores and changes in key EEG markers (Table 4), including Frontal Alpha Asymmetry (FAA), Beta Symmetry, and Theta/Alpha Ratios at F3 and F4. Although none of the correlations reached statistical significance, a modest negative trend emerged for the Theta/Alpha Ratio at F4 (r = −0.369, *p* = 0.194), suggesting that greater reductions in this ratio may be associated with more pronounced clinical improvement.

Changes in FAA, Beta Symmetry, and the Theta/Alpha Ratio at F3 exhibited minimal or weak correlations with clinical outcomes (all r < 0.30, *p* > 0.3). These findings support the interpretation that FAA and Beta Symmetry may represent more stable, trait-like aspects of cortical organization, whereas the Theta/Alpha Ratio—particularly at F4—may function as a more dynamic, state-dependent marker of treatment response. These results suggest that the frontal tDCS yields the most robust improvements in cognitive and motivational domains (especially concentration, sleep, and mood), with the most potent effects observed in individuals with moderate to severe depression (see Figure 4). 

This supports prior evidence that prefrontal neuromodulation is particularly effective in reversing executive dysfunction and motivational withdrawal (Table 5).

### 3.2. Impact of tDCS Treatment Intensity on Clinical Response

Correlation analyses were performed to examine the relationship between treatment exposure (total sessions and active weeks) and percentage change in MADRS scores across participants with mild and moderate/severe depression (Table 6).

In the mild depression group, a moderate negative correlation was found between the number of total sessions and MADRS percentage change (r = −0.49, *p* = 0.065), suggesting a trend toward greater clinical improvement with increased session exposure, although this did not reach statistical significance. Similarly, a moderate negative trend was observed between active treatment weeks and MADRS percentage change (r = −0.44, *p* = 0.099), which was also not statistically significant but indicative of a potential association.

In contrast, among participants with moderate to severe depression, correlations between treatment exposure and MADRS changes were weak and non-significant. For total sessions, the correlation was r = −0.19 (*p* = 0.50), while for active weeks, the correlation was r = −0.12 (*p* = 0.68).

These findings suggest that in mild depression, longer or more intensive treatment may be associated with greater symptom reduction, whereas in moderate to severe depression, clinical improvements occurred largely independent of the total number of sessions or duration of treatment.

Mild depression patients showed greater benefits with longer or more intensive tDCS courses, whereas moderate/severe patients achieved substantial improvements once a sufficient number of sessions was completed (Figure 5 and Figure 6).

### 3.3. Modulation of Frontal Alpha Asymmetry After tDCS

FAA was assessed before and after tDCS treatment to evaluate neurophysiological changes. In the mild depression group (*n* = 7), no significant change in FAA was detected (t (6) = 0.00, *p* = 1.00, Cohen’s d = 0.00), indicating no measurable impact on frontal asymmetry (Table 7).

In the moderate/severe depression group (*n* = 14), a small, non-significant reduction in FAA was observed (t (13) = −1.25, *p* = 0.233, Cohen’s d = −0.33). While the data suggested a modest trend toward normalization of frontal asymmetry, the change did not reach statistical significance.

In the mild depression group, no changes in FAA were observed following tDCS treatment (*p* = 1.00, Cohen’s d = 0), indicating no measurable neurophysiological effect.

In the moderate to severe depression group, a slight decrease in FAA was noted (Cohen’s d = −0.33), suggesting a modest trend toward normalization of frontal brain asymmetry. However, this change did not reach statistical significance (*p* = 0.233).

Overall, tDCS produced a more pronounced impact on clinical outcomes, as reflected by improvements in MADRS scores, than on FAA measures. In mild depression, clinical symptoms improved without corresponding EEG changes. In moderate to severe depression, a small, non-significant trend toward FAA normalization was observed alongside substantial clinical improvement.

### 3.4. Stability of Beta Symmetry Post-tDCS

In the mild depression group, no measurable changes in Beta Symmetry were observed (Table 8). Both FAA and Beta Symmetry remained stable throughout the course of tDCS treatment in these participants. Similarly, in the moderate to severe depression group, only a very small decrease in Beta Symmetry was detected (Cohen’s d = −0.15), which was not statistically significant (*p* = 0.587).

In the mild depression group, no significant changes in Theta/Alpha ratios were detected at either F3 (*p* = 0.586, Cohen’s d = −0.22) or F4 (*p* = 0.356, Cohen’s d = 0.38) (Table 9). These findings suggest that, consistent with other EEG measures, individuals with mild depression exhibited limited neurophysiological modulation despite experiencing clinical improvements.

In contrast, among participants with moderate to severe depression, a moderate reduction in the Theta/Alpha ratio at F3 was observed, approaching statistical significance (*p* = 0.080, Cohen’s d = −0.51). A significant decrease was identified at F4 (*p* = 0.018, Cohen’s d = −0.72), corresponding to a large effect size. This reduction is indicative of enhanced frontal cortical activation and may underlie the pronounced clinical improvements observed in this subgroup.

Our findings indicate that individuals with moderate to severe depression exhibited greater and more consistent improvements compared to those with mild symptoms, who showed only modest changes in certain areas or no improvement at all. These results suggest that tDCS may be more effective when baseline symptom severity is higher. Notably, certain symptoms—such as sleep disturbances and concentration difficulties—improved reliably in both groups, indicating symptom-specific responsiveness. This highlights the potential for tailoring tDCS protocols to target dominant symptom clusters for more personalized and effective interventions.

## 4. Discussion

This pilot study demonstrated that while tDCS treatment led to significant clinical improvements in depressive symptoms, its impact on Frontal Alpha Asymmetry (FAA) was limited. In the mild depression group, no measurable changes in FAA were observed post-treatment, suggesting that in individuals with milder baseline symptoms, tDCS may exert therapeutic effects without significantly altering frontal brain asymmetry. Among participants with moderate to severe depression, a small but non-significant decrease in FAA was detected (Cohen’s d = −0.33), reflecting a modest trend toward normalization of frontal cortical activity. However, this change did not achieve statistical significance.

These findings align with previous research suggesting that while FAA is a well-established biomarker of vulnerability to depression, it may not be highly sensitive to short-term neurophysiological changes induced by interventions such as tDCS [7,9]. Several factors may account for the limited FAA modulation observed: (1) FAA may function more as a stable trait marker rather than a dynamic state marker responsive to acute treatment; (2) individual variability in neuroplasticity, stimulation parameters, and montage choice may influence the degree of FAA change; and (3) resting-state EEG measures like FAA may not capture subtle or task-related neural alterations induced by tDCS.

Beta Symmetry, another EEG-based marker of frontal lobe function, was also assessed pre- and post-tDCS. In the mild depression group, Beta Symmetry remained stable following treatment, despite clinical improvements. Similarly, in the moderate to severe group, only a very small, non-significant reduction in Beta Symmetry was observed (Cohen’s d = −0.15, *p* = 0.587). These results suggest that Beta Symmetry, like FAA, may represent a relatively stable trait characteristic rather than a dynamic marker of treatment-induced neurophysiological change [14]. Although beta activity has been more strongly linked to anxiety and hypervigilance rather than core depressive symptoms [14], further research using task-evoked EEG paradigms or different tDCS montages may help clarify whether specific beta-related changes could emerge in subgroups of patients [8].

Although the left DLPFC (F3) served as the active stimulation site, F4 was also included—and ultimately emphasized—in our neurophysiological analysis due to its well-established involvement in emotion regulation, inhibitory control, and modulation of negative effects, all of which are commonly impaired in depression. Prior research indicates that depression is characterized not only by reduced activity in the left DLPFC but also by disrupted interhemispheric balance between the left and right prefrontal cortices. While individualized electric field modeling was not conducted in this study, existing evidence shows that standard tDCS montages produce diffuse rather than highly focal effects. Specifically, the F3 anode–F4 cathode configuration generates widespread current flow across frontal midline regions. Although F3 was the targeted stimulation site, the effects observed at F4 suggest that tDCS engages broader prefrontal networks beyond the local stimulation zone. These findings underscore the importance of lateralized EEG analysis and support the view that right prefrontal activity contributes meaningfully to recovery from depression, particularly in domains related to emotional regulation and cognitive disengagement.

Frontal theta activity (4–8 Hz) is commonly associated with cognitive-affective dysregulation. In the context of depression, elevated frontal theta—particularly in the prefrontal cortex—has been linked to cognitive slowing, reduced attentional control, and emotional disengagement. This increase in theta activity has also been associated with hypoactivity of the dorsolateral prefrontal cortex (DLPFC), a region essential for the top-down regulation of limbic-driven emotional responses [14,15].

Frontal alpha activity (8–12 Hz), especially in the high-alpha range, is typically regarded as a marker of cortical inhibition or “idling”. Elevated alpha power in the frontal regions—particularly in the right hemisphere—has been linked to avoidant emotional processing and reduced affective engagement [7,9].

When considered together, these two markers form the Theta/Alpha Ratio (TAR), a sensitive index of cortical disorganization and disengagement. Higher TAR values suggest impaired integration of cognitive and emotional processes, whereas reductions in this ratio, particularly following neuromodulation, may reflect re-engagement of prefrontal control circuits and the restoration of regulatory function.

While most tDCS studies targeting mood enhancement have focused on the left DLPFC (F3), the right DLPFC (F4) also plays a key role in emotional regulation, inhibitory control, and processing of negative effects. A reduction in TAR at F4 may, therefore, indicate improved cognitive control and decreased emotional dysregulation—changes that align with the clinical improvements observed in our study.

Previous research has shown that depression is not only characterized by hypoactivity in the left DLPFC, but also by disrupted interhemispheric balance between the left and right prefrontal cortices. Although F3 was the primary stimulation site in our protocol, the neurophysiological changes observed at F4 suggest that tDCS exerts effects beyond the targeted region, engaging broader prefrontal networks. These findings highlight the importance of lateralized EEG analysis and support the hypothesis that right-prefrontal involvement is mechanistically relevant to recovery from depression, particularly for symptoms related to emotional regulation and disengagement.

Theta/Beta ratios, commonly studied in ADHD and attentional regulation [16], were not a central focus of the present study, given their limited relevance as primary biomarkers of depression. Similarly, while the Theta/Alpha ratio has been associated with cognitive slowing in depression [15], it remains secondary rather than a core biomarker for mood pathology.

However, the Theta/Alpha ratio, particularly at the F4 electrode site, demonstrated promising sensitivity to tDCS intervention. In participants with moderate to severe depression, a significant reduction in the Theta/Alpha ratio at F4 was observed (*p* = 0.018, Cohen’s d = −0.72), indicating improved frontal cortical activation. This neurophysiological change likely reflects dynamic cortical reorganization underlying the robust clinical improvements seen in this group. These findings align with prior research emphasizing the potential of Theta/Alpha ratios as dynamic EEG biomarkers sensitive to neuromodulatory interventions [17,18].

The differing response patterns between the mild and moderate/severe depression groups suggest that baseline symptom severity may influence neurophysiological responsiveness to tDCS. These results highlight the potential value of EEG-informed stratification for personalizing tDCS protocols, allowing for optimization of stimulation parameters based on individual baseline characteristics.

Overall, while FAA and Beta Symmetry remained relatively unchanged, the observed normalization of Theta/Alpha ratios at F4 suggests that certain EEG metrics may more effectively capture short-term neurophysiological responses to tDCS, particularly in individuals with more severe depressive symptoms. This supports the potential role of Theta/Alpha ratios as biomarkers for tracking clinical response and guiding adaptive tDCS protocols in future research.

In this study, significant reductions in the Theta/Alpha Ratio at F4 were observed exclusively in participants with moderate to severe depression—the group that also exhibited the greatest clinical improvement. This pattern suggests that elevated prefrontal Theta/Alpha ratios may represent a neurophysiological signature of the “depressed state” and could serve as a baseline marker for identifying individuals most likely to benefit from tDCS. In clinical settings, pretreatment EEG screening may facilitate patient stratification by prioritizing tDCS for those with elevated Theta/Alpha activity, while guiding others toward alternative or adjunctive interventions when ratios are already within normal ranges. Moreover, because the Theta/Alpha Ratio decreased in parallel with symptom improvement, it may also function as a dynamic biomarker of treatment response. Mid-treatment monitoring of this ratio could help clinicians determine whether to maintain, intensify, or adjust the intervention based on early neurophysiological changes.

## 5. Conclusions

This study found that frontal lobe tDCS was associated with significant clinical self-reported improvements in patients with both mild and moderate to severe depression, as reflected by reductions in MADRS scores. Notably, participants with reported moderate to severe baseline symptoms exhibited higher clinical response rates compared to those with milder symptoms.

Although no significant changes were detected in Frontal Alpha Asymmetry (FAA) or Beta Symmetry following tDCS, a promising decrease in the Theta/Alpha Ratio at F4 was observed among participants with moderate to severe depression. This finding suggests that enhanced frontal cortical activation may serve as a potential neurophysiological indicator of treatment response.

Several important conclusions emerge from these results. First, tDCS proves to be an effective clinical intervention for alleviating depressive symptoms, particularly in individuals with more severe forms of the disorder. Second, meaningful clinical improvements may occur even in the absence of large-scale changes in resting-state EEG metrics. Third, the Theta/Alpha Ratio at F4 presents itself as a potential dynamic marker of tDCS-induced cortical plasticity in the treatment of depression.

Future studies should aim to investigate these findings in larger and more homogeneous samples, explore the benefits of individualized tDCS protocols, employ more sensitive neurophysiological techniques (such as functional connectivity analyses and task-based EEG paradigms), and assess long-term clinical and neurophysiological outcomes.

## 6. Limitations

Despite the encouraging clinical outcomes, several limitations should be acknowledged. The most significant is the absence of a sham control group, which limits our ability to draw definitive causal inferences regarding the effects of tDCS. Without a blinded, placebo-controlled condition, observed symptom improvements may be influenced by expectancy effects or natural symptom fluctuation. Additionally, reliance on self-reported measures such as the MADRS—though widely used in clinical research—introduces potential bias related to participant expectations. To more accurately assess the efficacy and specificity of tDCS, future studies should incorporate double-blind, sham-controlled designs.

The relatively small sample size—especially after stratifying participants by baseline depression severity—limits the statistical power of our analyses and restricts the generalizability of the findings. Consequently, results should be considered preliminary. Larger, adequately powered studies are needed to replicate these findings and further explore the moderating role of baseline symptom severity. Additionally, participant heterogeneity in demographic characteristics, comorbid conditions, and treatment histories may have introduced confounding factors that were not fully controlled for.

Neurophysiological assessments were limited to resting-state EEG markers (Frontal Alpha Asymmetry, Beta Symmetry, and Theta/Alpha Ratio). Other potentially informative techniques—such as functional connectivity analysis, task-evoked EEG, or source localization—were not utilized [19,20]. Furthermore, outcomes were evaluated only immediately post-intervention, with no long-term follow-up to assess the durability of clinical or neurophysiological changes.

While the use of a fixed tDCS montage and stimulation parameters ensured experimental consistency, this approach may not have been optimal for all individuals, given neurobiological variability [21,22]. Finally, the number of statistical comparisons conducted across clinical and EEG measures increases the risk of both Type I and Type II errors, particularly in the context of a limited sample size.

Future research should aim to address these limitations by employing larger, more homogeneous samples, personalized stimulation protocols, and more sensitive neurophysiological tools—including functional near-infrared spectroscopy (fNIRS) [23]. Longitudinal study designs will also be critical for capturing the trajectory and persistence of both clinical and neural changes. Further investigation into whether changes in Beta Symmetry or Theta/Alpha Ratio predict long-term remission or relapse prevention, and the integration of EEG-based connectivity or coherence measures, may enhance the precision of biomarkers for tDCS efficacy in depression.

Despite certain limitations, this pilot study adds valuable evidence supporting the clinical efficacy of tDCS and suggests that emerging EEG biomarkers could play a crucial role in personalizing neuromodulatory interventions for depressio.

## Figures and Tables

**Figure 1 brainsci-15-00540-f001:**
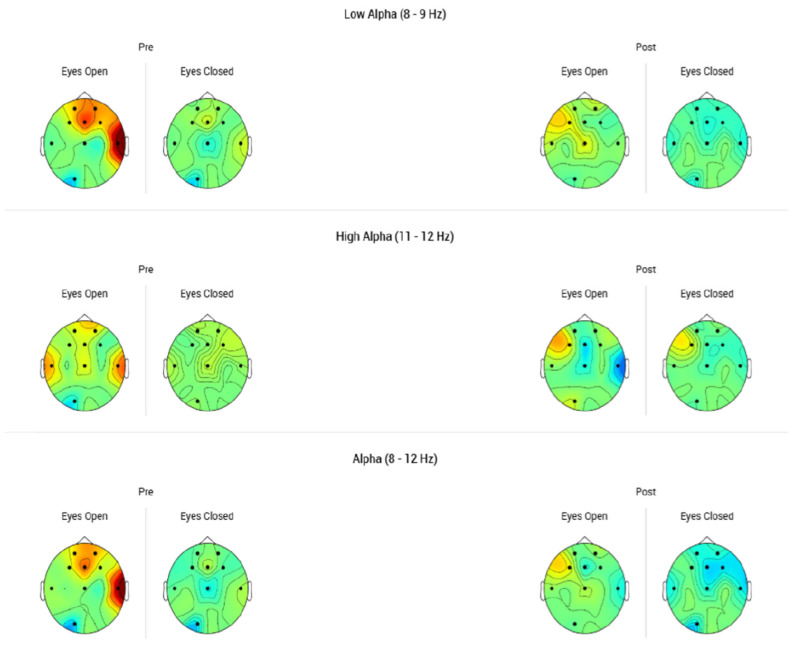
sQEEG measured before and after tDCS treatment, comparing the Alpha waves, performed with the Myndlift app and the Muse 2 headband, showing a trend towards normalized values. Voltage (µV) for each electrode relative to the mean voltage for the normative population in the same age group on the date of the first assessment. Red color represents a higher amplitude than the Z database; blue color represents a lower amplitude than the Z database. Increments of color are standardized units of standard deviation with 0 indicating no difference compared to the database. The black dots represent the electrode placement during the EEG recordings, according to the 10–20 international EEG standard system.

**Figure 2 brainsci-15-00540-f002:**
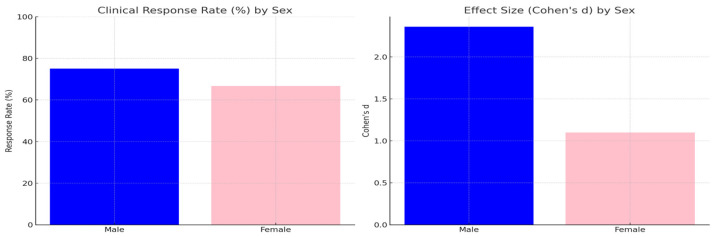
Sex-based differences in clinical outcomes following tDCS. Left panel: Clinical response rate (%) by sex. Right panel: Effect size (Cohen’s d) by sex. Both groups demonstrated significant improvements, with males showing slightly greater clinical benefits.

**Figure 3 brainsci-15-00540-f003:**
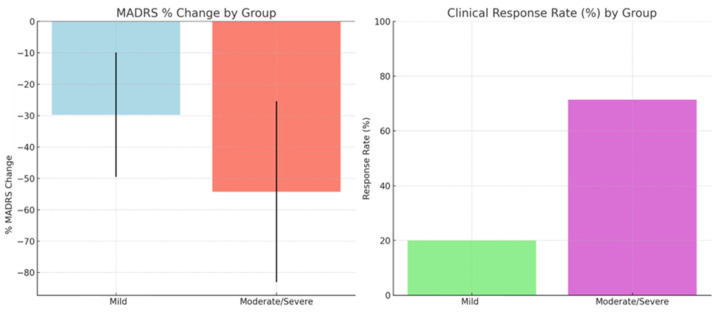
Comparison of MADRS score reductions and clinical response rates between participants with mild depression and those with moderate to severe depression following tDCS treatment.

**Figure 4 brainsci-15-00540-f004:**
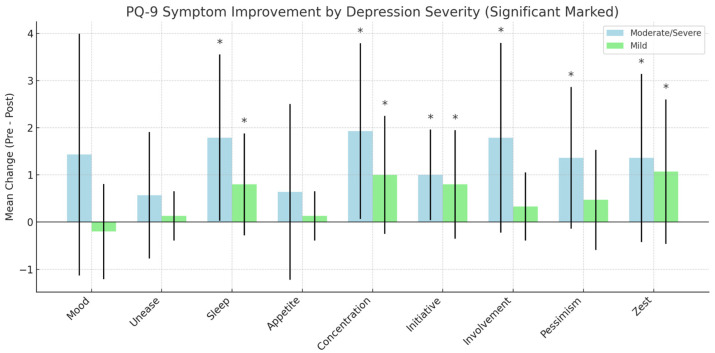
PHQ–9 symptom improvement between participants with Moderate/Severe and Mild depression: following tDCS treatment. Note: * = significant improvement.

**Figure 5 brainsci-15-00540-f005:**
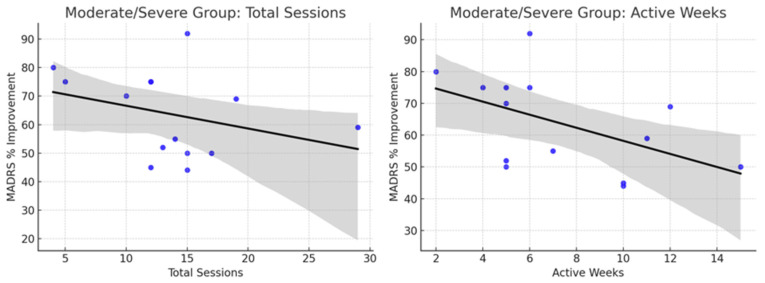
Scatterplot illustrating the relationship between total sessions, active treatment weeks, and percentage improvement in MADRS scores among participants with baseline mild depression. Note: Grey line represents standard deviation, Black line represents trendline and Dots represent the intersection between MARDS % improvement vs. Total sessions, or MARDS % improvements vs. Active weeks. More sessions and more active weeks tend to correlate with greater reductions in MADRS scores. The trendlines are downward, meaning more treatment more symptom improvement.

**Figure 6 brainsci-15-00540-f006:**
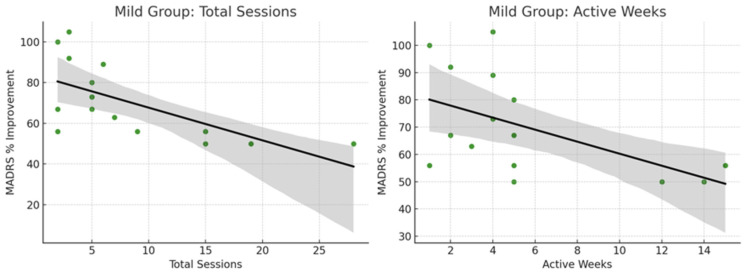
A scatterplot illustrates the relationship between total sessions, active treatment weeks, and percentage improvement in MADRS scores among participants with baseline moderate to severe depression.

**Table 1 brainsci-15-00540-t001:** Statistical analysis of clinical outcomes in participants with moderate to severe depression (MADRS score ≥ 20) following tDCS treatment.

Analysis	Result
Paired *t*–test (MADRS Start vs. End)	*t* (13) = 5.87, *p* = 0.000055 (highly significant)
Cohen’s d (effect size)	1.57 (very large effect)
Clinical Response Rate	71.4% (10 out of 14 patients achieved ≥50% reduction)

**Table 2 brainsci-15-00540-t002:** Sex-based subgroup analysis of clinical outcomes following tDCS treatment.

Sex	*n*	t-Statistic	*p*-Value	Cohen’s d	Clinical Response Rate (%)
Male	8	6.68	0.00028	2.36	75.0%
Female	6	2.70	0.043	1.10	66.7%

**Table 3 brainsci-15-00540-t003:** Clinical outcomes for participants with mild depression (baseline MADRS scores between 7 and 19) following tDCS treatment.

Analysis	Result
Paired *t*–test (MADRS Start vs. End)	t(14) = 5.41, *p* = 0.000091 (highly significant)
Cohen’s d (effect size)	1.40 (very large effect)
Clinical Response Rate	20% (3 out of 15 patients achieved ≥50% reduction)

**Table 4 brainsci-15-00540-t004:** Correlation between MADRS improvement and EEG Change.

Neurophysiological Marker	Pearson *r*	*p*-Value
FAA	0.293	0.3101
BetaSym	−0.144	0.6229
ThetaAlphaF3	−0.072	0.8058
ThetaAlphaF4	−0.369	0.1943

**Table 5 brainsci-15-00540-t005:** Statistically Significant Improvements: following tDCS treatment.

Symptom	Mean Change	*p*-Value	Cohen’s d	Interpretation
Sleep	1.79	0.0022	1.01	Large improvement
Concentration	1.93	0.0019	1.04	Large improvement
Mood	1.43	0.0573	0.56	Moderate effect
Unease	0.57	0.1352	0.43	Small–moderate
Appetite	0.64	0.2196	0.34	Small

**Table 6 brainsci-15-00540-t006:** Correlation analysis between treatment exposure (total sessions and active weeks) and percentage change in MADRS scores for participants with mild and moderate/severe depression.

Group	Measure	Correlation (r)	*p*-Value	Interpretation
Mild	Total Sessions vs. MADRS% Change	r = −0.49	0.065	Moderate negative trend, nearly significant
Mild	Active Weeks vs. MADRS% Change	r = −0.44	0.099	Moderate negative trend, not significant
Moderate/Severe	Total Sessions vs. MADRS% Change	r = −0.19	0.50	Weak, non-significant
Moderate/Severe	Active Weeks vs. MADRS% Change	r = −0.12	0.68	Very weak, non-significant

**Table 7 brainsci-15-00540-t007:** Changes in frontal alpha asymmetry pre- and post-tDCS treatment for participants with mild and moderate/severe depression.

Group	*n*	Paired *t*-Test (*p*-Value)	Cohen’s d	Interpretation
Mild	7	t (6) = 0.00, *p* = 1.00	0.00	No change at all
Moderate/Severe	14	t (13) = −1.25, *p* = 0.233	−0.33	Small improvement, not significant

**Table 8 brainsci-15-00540-t008:** Beta Symmetry changes pre- and post-tDCS treatment for participants with mild and moderate/severe depression.

Group	*n*	Paired *t*-Test (*p*-Value)	Cohen’s d	Interpretation
Mild	7	NaN	NaN	No variation detected
Moderate/Severe	14	t (13) = −0.56, *p* = 0.587	−0.15	Very small, non-significant change

**Table 9 brainsci-15-00540-t009:** Changes in Theta/Alpha ratios at F3 and F4 pre- and post-tDCS treatment for participants with mild and moderate/severe depression.

Group	Location	*n*	Paired *t*-Test (*p*-Value)	Cohen’s d	Interpretation
Mild	F3	7	*p* = 0.586	−0.22	No significant change
Mild	F4	7	*p* = 0.356	+0.38	No significant change
Moderate/Severe	F3	14	*p* = 0.080	−0.51	Trend toward significance (moderate effect)
Moderate/Severe Group	F4	14	*p* = 0.018	−0.72	Significant decrease (large effect)

## Data Availability

The original contributions presented in this study are included in the article; further inquiries can be directed to the corresponding author.

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
