# Peer review of "Frontal Transcranial Direct Current Stimulation in Moderate to Severe Depression: Clinical and Neurophysiological Findings from a Pilot Study"

_brainsci, 2025, doi:10.3390/brainsci15060540_

Round 1
Reviewer 1 Report
Comments and Suggestions for Authors
This pilot study investigates the clinical and neurophysiological effects of frontal tDCS in individuals with mild to severe depression, focusing on symptom reduction via MADRS scores and EEG markers, including FAA, Beta Symmetry, and Theta/Alpha ratios. The results demonstrate significant clinical improvements, particularly in moderate/severe cases, though neurophysiological changes were limited to a reduction in the Theta/Alpha ratio at F4, suggesting enhanced right frontal activation. While FAA and Beta Symmetry remained stable, the findings imply that tDCS may alleviate depressive symptoms without broadly altering resting-state EEG asymmetry. However, methodological limitations, such as small sample sizes, lack of sham control, and reliance on consumer-grade EEG, warrant cautious interpretation. More detailed comments are listed below:
- The inclusion of participants with baseline MADRS scores below 6 is problematic. These individuals do not align with the study's focus on depression and may confound results. From my point of view, the rationale for their inclusion should be clarified, or their data should be excluded.
- The significant reduction in the Theta/Alpha Ratio at F4 is highlighted as a key finding, but the physiological interpretation is oversimplified. Please elaborate on why this ratio, particularly at F4, is mechanistically relevant to tDCS effects in depression, linking it to previous works on frontal cortical activation and cognitive-emotional processing.
- The significant reduction in the Theta/Alpha Ratio at F4 is intriguing, but why was F4prioritized over F3? Given the anode’s placement at F3, was any asymmetry in tDCS-induced electric fields modeled to interpret lateralized effects?
- The use of a 50% reduction in MADRS scores as the sole criterion for clinical response is arbitrary. It is necessary to justify this threshold and fully consider complementary metrics, such as remission rates and effect sizes to provide a more detailed evaluation of clinical efficacy.
- The limitations section overlooks critical issues, such as the absence of a sham control group, which undermines causal inferences about tDCS effects. Additionally, the potential for placebo effects due to self-reported outcomes should be discussed.
- The study uses a fixed 2 mA, 30-minute tDCS protocol, yet individual differences in skull thickness, electrode contact, and neural responsivity may influence outcomes. Was impedance monitored during sessions to ensure consistent current delivery?
- Finally, the Flow app included psychotherapy-based training modules. Were these standardized across participants? If not, their content and usage frequency should be reported, as they may independently influence mood and EEG measures.
Author Response
Manuscript Number: brainsci-3649588
Dear editorial office,
We are pleased to submit the revised version of our manuscript, entitled “Frontal transcranial direct current stimulation in moderate to severe depression: Clinical and neurophysiological findings from a pilot study” (Manuscript ID: brainsci-3649588), for your consideration in Brain Sciences.
We have thoroughly reviewed the comments provided by the reviewers and have made significant revisions to address their suggestions and concerns. A detailed, point-by-point response to each comment is included below. To facilitate the review process, all major changes in the manuscript are highlighted in blue. Additionally, the entire manuscript has undergone English language proofreading.
Thank you for considering our revised submission. We look forward to your feedback. For any further correspondence regarding this manuscript, please contact me at Cristina.dumitru81@upb.ro.
Yours sincerely,
Cristina Dumitru
Reviewer #1:
First, we would like to thank you for carefully reviewing and considering our paper. We appreciate the feedback and hope we improved our manuscript significantly.
Point 1. The inclusion of participants with baseline MADRS scores below 6 is problematic. These individuals do not align with the study's focus on depression and may confound results. From my point of view, the rationale for their inclusion should be clarified, or their data should be excluded.
Response 1
Thank you for your suggestion. We agree that MADRS scores below 6 generally reflect an absence of clinically significant depressive symptoms, and we understand the potential concern regarding their inclusion in a study focused on depression.
To address this, we have included the following clarifications in the manuscript:
Participants were enrolled in an open-label, real-world pilot study investigating the effects of a standardized tDCS protocol delivered via a commercially available medical device (Flow Neuroscience). Accordingly, no pre-screening based on MADRS score thresholds was conducted at intake, reflecting naturalistic usage patterns in early-intervention contexts where individuals may seek neuromodulation for subthreshold symptoms or relapse prevention. To reduce the potential confounding effect of very low MADRS scores, we structured our analyses to distinguish between participants with moderate to severe depression (MADRS ≥20) and those with mild symptoms (MADRS 7–19). Two participants with MADRS scores <6 were excluded from the clinical response rate analysis of the mild group (as shown in Table 3), and their data were not included in outcome comparisons across depressive severity groups.
Point 2. The significant reduction in the Theta/Alpha Ratio at F4 is highlighted as a key finding, but the physiological interpretation is oversimplified. Please elaborate on why this ratio, particularly at F4, is mechanistically relevant to tDCS effects in depression, linking it to previous works on frontal cortical activation and cognitive-emotional processing.
Response 2
We acknowledge that our initial interpretation of the reduction in the Theta/Alpha Ratio (TAR) at F4 may have been overly simplistic, and we appreciate the opportunity to provide a more detailed explanation of its physiological significance in the context of transcranial direct current stimulation (tDCS) and depression.
We added: “Frontal theta activity (4–8 Hz) is commonly associated with cognitive-affective dysregulation. In the context of depression, elevated frontal theta—particularly in the prefrontal cortex—has been linked to cognitive slowing, reduced attentional control, and emotional disengagement. This increase in theta activity has also been associated with hypoactivity of the dorsolateral prefrontal cortex (DLPFC), a region essential for the top-down regulation of limbic-driven emotional responses [14, 17].
Frontal alpha activity (8–12 Hz), especially in the high-alpha range, is typically regarded as a marker of cortical inhibition or "idling." Elevated alpha power in the frontal regions—particularly in the right hemisphere—has been linked to avoidant emotional processing and reduced affective engagement [8, 10].
When considered together, these two markers form the Theta/Alpha Ratio (TAR), a sensitive index of cortical disorganization and disengagement. Higher TAR values suggest impaired integration of cognitive and emotional processes, whereas reductions in this ratio, particularly following neuromodulation, may reflect re-engagement of prefrontal control circuits and the restoration of regulatory function.
While most tDCS studies targeting mood enhancement have focused on the left DLPFC (F3), the right DLPFC (F4) also plays a key role in emotional regulation, inhibitory control, and processing of negative affect. A reduction in TAR at F4 may therefore indicate improved cognitive control and decreased emotional dysregulation—changes that align with the clinical improvements observed in our study.
Previous research has shown that depression is not only characterized by hypoactivity in the left DLPFC, but also by disrupted interhemispheric balance between the left and right prefrontal cortices. Although F3 was the primary stimulation site in our protocol, the neurophysiological changes observed at F4 suggest that tDCS exerts effects beyond the targeted region, engaging broader prefrontal networks. These findings highlight the importance of lateralized EEG analysis and support the hypothesis that right-prefrontal involvement is mechanistically relevant to recovery from depression, particularly for symptoms related to emotional regulation and disengagement.”
Point 3. The significant reduction in the Theta/Alpha Ratio at F4 is intriguing, but why was F4prioritized over F3? Given the anode’s placement at F3, was any asymmetry in tDCS-induced electric fields modeled to interpret lateralized effects?
Response 3
Thank you for this important observation. While the anodal electrode was indeed placed at F3 (left DLPFC), we did not explicitly model the electric field distribution across hemispheres in this pilot study. However, the rationale for highlighting changes at F4 rather than F3 is based on both the observed neurophysiological effect and the known functional contributions of the right DLPFC to emotion regulation and inhibitory control.
To address this, in our manuscript we have included:
“Although the left DLPFC (F3) served as the active stimulation site, F4 was also included—and ultimately emphasized—in our neurophysiological analysis due to its well-established involvement in emotion regulation, inhibitory control, and modulation of negative affect, all of which are commonly impaired in depression. Prior research indicates that depression is characterized not only by reduced activity in the left DLPFC but also by disrupted interhemispheric balance between the left and right prefrontal cortices. While individualized electric field modeling was not conducted in this study, existing evidence shows that standard tDCS montages produce diffuse rather than highly focal effects. Specifically, the F3 anode–F4 cathode configuration generates widespread current flow across frontal midline regions. Although F3 was the targeted stimulation site, the effects observed at F4 suggest that tDCS engages broader prefrontal networks beyond the local stimulation zone. These findings underscore the importance of lateralized EEG analysis and support the view that right prefrontal activity contributes meaningfully to recovery from depression, particularly in domains related to emotional regulation and cognitive disengagement.”
Point 4. The use of a 50% reduction in MADRS scores as the sole criterion for clinical response is arbitrary. It is necessary to justify this threshold and fully consider complementary metrics, such as remission rates and effect sizes to provide a more detailed evaluation of clinical efficacy.
Response 4
We have clarified by adding:
“Defining treatment response as a 50% reduction in MADRS scores aligns with FDA and EMA guidelines and is a widely accepted standard in both pharmacological and neuromodulation research, including studies on rTMS, ECT, and tDCS.”
Point 5. The limitations section overlooks critical issues, such as the absence of a sham control group, which undermines causal inferences about tDCS effects. Additionally, the potential for placebo effects due to self-reported outcomes should be discussed.
Response 5
We fully agree with the reviewer’s comment that the absence of a sham control group represents a significant limitation that affects the strength of causal inferences regarding the efficacy of tDCS in our study. While our findings suggest a beneficial effect, we acknowledge that without a placebo-controlled condition, we cannot rule out the possibility that observed improvements were influenced by non-specific factors such as participant expectations or the natural course of symptoms.
We have now revised the Limitations section and added:
“Despite the encouraging clinical outcomes, several limitations should be acknowledged. The most significant is the absence of a sham control group, which limits our ability to draw definitive causal inferences regarding the effects of tDCS. Without a blinded, placebo-controlled condition, observed symptom improvements may be influenced by expectancy effects or natural symptom fluctuation. Additionally, reliance on self-reported measures such as the MADRS—though widely used in clinical research—introduces potential bias related to participant expectations. To more accurately assess the efficacy and specificity of tDCS, future studies should incorporate double-blind, sham-controlled designs.”
Point 6. The study uses a fixed 2 mA, 30-minute tDCS protocol, yet individual differences in skull thickness, electrode contact, and neural responsivity may influence outcomes. Was impedance monitored during sessions to ensure consistent current delivery?
Response 6
We have now added this clarification in the Methods section under the tDCS procedure:
“Impedance was monitored in real time before and throughout each tDCS session using the device’s built-in impedance check feature. Stimulation commenced only when impedance levels were below the safety thresholds specified by the manufacturer. Continuous monitoring ensured stable electrode contact during the session, and electrodes were readjusted if impedance levels rose during stimulation.”
Point 7. Finally, the Flow app included psychotherapy-based training modules. Were these standardized across participants? If not, their content and usage frequency should be reported, as they may independently influence mood and EEG measures.
Response 7
We did not use the psychotherapy-based training of the Flow app.
We would like to thank you again for your efforts in reviewing our manuscript and we are grateful for this opportunity to refine and strengthen our manuscript.
Reviewer 2 Report
Comments and Suggestions for Authors
This pilot study makes a valuable contribution to the field by examining both clinical outcomes and neurophysiological correlates of transcranial direct current stimulation (tDCS) for depression. The investigation of Frontal Alpha Asymmetry (FAA), Beta Symmetry, and Theta/Alpha Ratios as potential biomarkers is particularly noteworthy. The manuscript is generally well-structured, but would benefit from several improvements before publication.
Specific Comments
Strengths:
- The differentiation between mild and moderate/severe depression groups provides important insights into varying treatment efficacy across severity levels.
- The study makes a compelling case for the Theta/Alpha Ratio as a potential biomarker for tDCS response in depression.
- The comprehensive assessment of both clinical (MADRS, PHQ-9) and neurophysiological measures offers a multidimensional perspective on treatment effects.
- The statistical analysis is appropriate, with effect sizes reported alongside significance values.
Areas for Improvement:
Methodology:
- The sample size (n=31) is modest, with further subdivision into mild and moderate/severe groups limiting statistical power. While appropriate for a pilot study, this limitation should be more prominently acknowledged.
- The use of the Muse 2 headband for EEG assessment, while practical, has limitations in spatial resolution and signal quality compared to clinical-grade EEG systems. More discussion of these technical limitations would strengthen the paper.
- The relationship between the "PQ-9" mentioned in results (Table 7) and the "PHQ-9" described in methods requires clarification - these appear to be the same measure but are inconsistently labeled.
- More details on participant characteristics would be valuable, including treatment history, medication status, and comorbidities.
Results Presentation:
- Table 8 appears incomplete, missing data for F4 in the moderate/severe group despite this being highlighted as a significant finding in the text.
- Some figures would benefit from improved axis labeling and clearer titles.
- The scatter plots (Figures 5 and 6) could include trend lines to better illustrate the correlational relationships described.
Discussion:
- The discussion of sex differences in treatment response is interesting but somewhat underdeveloped. Given the significant differences observed (Cohen's d = 2.36 for males vs. 1.10 for females), this warrants more extensive exploration.
- The practical clinical implications of the findings, particularly regarding the potential use of the Theta/Alpha ratio as a biomarker, could be further elaborated.
- Consider discussing how these findings might inform personalized tDCS protocols, especially given the different response patterns between mild and moderate/severe depression groups.
Editorial Issues:
- Several typographical errors need correction:
- Line 223: "Tese" should be "These"
- Line 229: "specialy" should be "especially"
- Inconsistent use of PQ-9/PHQ-9 terminology
- Reference formatting is inconsistent throughout the manuscript.
- The abbreviations list on page 12 mentions "PQ-9" as potentially a mislabel of "PHQ-9" - this needs resolution.
Recommendations
- Address the terminology inconsistencies, particularly regarding PHQ-9/PQ-9.
- Complete Table 8 to include the missing F4 data for the moderate/severe group.
- Enhance the discussion of sex differences and their potential clinical implications.
- Expand on how the Theta/Alpha ratio findings might be translated into clinical practice.
- Consider including a correlation analysis between clinical improvements and neurophysiological changes.
- Improve figure labeling and presentation for clarity.
- Correct typographical and grammatical errors throughout the manuscript.
This study has significant potential to contribute to our understanding of tDCS mechanisms in depression treatment. With the suggested revisions, it would make a stronger contribution to the literature and provide valuable direction for future research in this important area.
Author Response
Manuscript Number: brainsci-3649588
Dear editorial office,
We are pleased to submit the revised version of our manuscript, entitled “Frontal transcranial direct current stimulation in moderate to severe depression: Clinical and neurophysiological findings from a pilot study” (Manuscript ID: brainsci-3649588), for your consideration in Brain Sciences.
We have thoroughly reviewed the comments provided by the reviewers and have made significant revisions to address their suggestions and concerns. A detailed, point-by-point response to each comment is included below. To facilitate the review process, all major changes in the manuscript are highlighted in blue. Additionally, the entire manuscript has undergone English language proofreading.
Thank you for considering our revised submission. We look forward to your feedback. For any further correspondence regarding this manuscript, please contact me at Cristina.dumitru81@upb.ro.
Yours sincerely,
Cristina Dumitru
Reviewer #2:
Point 1. The sample size (n=31) is modest, with further subdivision into mild and moderate/severe groups limiting statistical power. While appropriate for a pilot study, this limitation should be more prominently acknowledged.
Response 1.
We appreciate the reviewers’ observation regarding the modest sample size (n=31) and the subsequent subdivision into mild and moderate/severe depression groups, which indeed limits the statistical power and generalizability of the findings. In response, emphasized this limitation more explicitly in the revised Limitations section:
“The relatively small sample size—especially after stratifying participants by baseline depression severity—limits the statistical power of our analyses and restricts the generalizability of the findings. Consequently, results should be considered preliminary. Larger, adequately powered studies are needed to replicate these findings and further explore the moderating role of baseline symptom severity. Additionally, participant heterogeneity in demographic characteristics, comorbid conditions, and treatment histories may have introduced confounding factors that were not fully controlled for.”
Point 2. The use of the Muse 2 headband for EEG assessment, while practical, has limitations in spatial resolution and signal quality compared to clinical-grade EEG systems. More discussion of these technical limitations would strengthen the paper.
Response 2.
We are grateful for the reviewer's observations. We included the following:
“The aim of our study was to develop an accessible, user-friendly tool for practitioners, with a specific focus on differences in activity between the left and right frontal hemispheres—given that individuals with depression typically exhibit relatively reduced left compared to right resting frontal activity (Stewart et al., 2012). Despite its limited number of electrodes, the Muse device has demonstrated the ability to effectively distinguish between high and low valence/arousal emotional states, with an accuracy comparable to that achieved using the full set of DEAP electrodes.”
Point 3. The relationship between the "PQ-9" mentioned in results (Table 7) and the "PHQ-9" described in methods requires clarification - these appear to be the same measure but are inconsistently labeled.
Response 3.
Thank you for your pertinent observation, we have checked the manuscript and corrected with “PHQ-9”.
Point 4. More details on participant characteristics would be valuable, including treatment history, medication status, and comorbidities.
Response 4.
Thank you for your comment. We have added more information on participant characteristics, as follows:
“In addition to age and baseline depression severity, clinical background information was collected. Eighteen participants (58%) reported a history of treatment for depression, and ten (32%) were on stable pharmacological therapy during the study—primarily selective serotonin reuptake inhibitors. Medication regimens were maintained without changes throughout the intervention period. Comorbid psychiatric conditions included anxiety disorders (n = 6), mild attention-deficit/hyperactivity disorder (n = 2), and somatic symptom presentations (n = 3). These factors were not used as exclusion criteria in order to maintain ecological validity, but they were documented and taken into account during data interpretation.”
Point 5. Table 8 appears incomplete, missing data for F4 in the moderate/severe group despite this being highlighted as a significant finding in the text.
Response 5.
Thank you for your pertinent observation, indeed we missed a line in table 8, we have now added.
Point 6. Some figures would benefit from improved axis labeling and clearer titles.
Response 6.
Thank you for your recommendation, we have checked and refined our Figures.
Point 7. The scatter plots (Figures 5 and 6) could include trend lines to better illustrate the correlational relationships described.
Response 7.
We appreciate the reviewer’s feedback and understand the importance of clear reporting so we modified figures, to be clearer.
Point 8. The discussion of sex differences in treatment response is interesting but somewhat underdeveloped. Given the significant differences observed (Cohen's d = 2.36 for males vs. 1.10 for females), this warrants more extensive exploration.
Response 8.
Thank you for pointing it out, we have added the following paragraph to Result section.
“While both male and female participants showed significant improvements following tDCS, the effect size was notably greater in males. This may point to sex-specific differences in neuromodulatory responsiveness or in the neural circuits involved in emotion regulation targeted by DLPFC stimulation. However, these findings are preliminary and should be interpreted with caution. Further research with larger sample sizes is needed to better understand the potential moderating role of sex in tDCS treatment outcomes.”
Point 8. The practical clinical implications of the findings, particularly regarding the potential use of the Theta/Alpha ratio as a biomarker, could be further elaborated.
Response 8. Thank you for your insightful feedback. We have revised the discussion section, and we added the following:
“Although the left DLPFC (F3) served as the active stimulation site, F4 was also included—and ultimately emphasized—in our neurophysiological analysis due to its well-established involvement in emotion regulation, inhibitory control, and modulation of negative affect, all of which are commonly impaired in depression. Prior research indicates that depression is characterized not only by reduced activity in the left DLPFC but also by disrupted interhemispheric balance between the left and right prefrontal cortices. While individualized electric field modeling was not conducted in this study, existing evidence shows that standard tDCS montages produce diffuse rather than highly focal effects. Specifically, the F3 anode–F4 cathode configuration generates widespread current flow across frontal midline regions. Although F3 was the targeted stimulation site, the effects observed at F4 suggest that tDCS engages broader prefrontal networks beyond the local stimulation zone. These findings underscore the importance of lateralized EEG analysis and support the view that right prefrontal activity contributes meaningfully to recovery from depression, particularly in domains related to emotional regulation and cognitive disengagement.
Frontal theta activity (4–8 Hz) is commonly associated with cognitive-affective dysregulation. In the context of depression, elevated frontal theta—particularly in the prefrontal cortex—has been linked to cognitive slowing, reduced attentional control, and emotional disengagement. This increase in theta activity has also been associated with hypoactivity of the dorsolateral prefrontal cortex (DLPFC), a region essential for the top-down regulation of limbic-driven emotional responses [14, 17].
Frontal alpha activity (8–12 Hz), especially in the high-alpha range, is typically regarded as a marker of cortical inhibition or "idling." Elevated alpha power in the frontal regions—particularly in the right hemisphere—has been linked to avoidant emotional processing and reduced affective engagement [8, 10].
When considered together, these two markers form the Theta/Alpha Ratio (TAR), a sensitive index of cortical disorganization and disengagement. Higher TAR values suggest impaired integration of cognitive and emotional processes, whereas reductions in this ratio, particularly following neuromodulation, may reflect re-engagement of prefrontal control circuits and the restoration of regulatory function.
While most tDCS studies targeting mood enhancement have focused on the left DLPFC (F3), the right DLPFC (F4) also plays a key role in emotional regulation, inhibitory control, and processing of negative affect. A reduction in TAR at F4 may therefore indicate improved cognitive control and decreased emotional dysregulation—changes that align with the clinical improvements observed in our study.
Previous research has shown that depression is not only characterized by hypoactivity in the left DLPFC, but also by disrupted interhemispheric balance between the left and right prefrontal cortices. Although F3 was the primary stimulation site in our protocol, the neurophysiological changes observed at F4 suggest that tDCS exerts effects beyond the targeted region, engaging broader prefrontal networks. These findings highlight the importance of lateralized EEG analysis and support the hypothesis that right-prefrontal involvement is mechanistically relevant to recovery from depression, particularly for symptoms related to emotional regulation and disengagement.”
Point 9. Consider discussing how these findings might inform personalized tDCS protocols, especially given the different response patterns between mild and moderate/severe depression groups.
Response 9. Based on your recommendations, we added:
“The differing response patterns between the mild and moderate/severe depression groups suggest that baseline symptom severity may influence neurophysiological responsiveness to tDCS. These results highlight the potential value of EEG-informed stratification for personalizing tDCS protocols, allowing for optimization of stimulation parameters based on individual baseline characteristics.”
Point 10. Consider discussing how these findings might inform personalized tDCS protocols, especially given the different response patterns between mild and moderate/severe depression groups.
Response 10. We have added:
“Our findings indicate that individuals with moderate to severe depression exhibited greater and more consistent improvements compared to those with mild symptoms, who showed only modest changes in certain areas or no improvement at all. These results suggest that tDCS may be more effective when baseline symptom severity is higher. Notably, certain symptoms—such as sleep disturbances and concentration difficulties—improved reliably in both groups, indicating symptom-specific responsiveness. This highlights the potential for tailoring tDCS protocols to target dominant symptom clusters for more personalized and effective interventions.”
Point 11. Editorial Issues:
- Several typographical errors need correction:
- Line 223: "Tese" should be "These"
- Line 229: "specialy" should be "especially"
- Inconsistent use of PQ-9/PHQ-9 terminology
- Reference formatting is inconsistent throughout the manuscript.
- The abbreviations list on page 12 mentions "PQ-9" as potentially a mislabel of "PHQ-9" - this needs resolution.
Response 11. We have reviewed according to your comments and recommendations.
Point 12. Expand on how the Theta/Alpha ratio findings might be translated into clinical practice.
Response 12. We have added:
“In this study, significant reductions in the Theta/Alpha Ratio at F4 were observed exclusively in participants with moderate to severe depression—the group that also exhibited the greatest clinical improvement. This pattern suggests that elevated prefrontal Theta/Alpha ratios may represent a neurophysiological signature of the “depressed state” and could serve as a baseline marker for identifying individuals most likely to benefit from tDCS. In clinical settings, pretreatment EEG screening may facilitate patient stratification by prioritizing tDCS for those with elevated Theta/Alpha activity, while guiding others toward alternative or adjunctive interventions when ratios are already within normal ranges. Moreover, because the Theta/Alpha Ratio decreased in parallel with symptom improvement, it may also function as a dynamic biomarker of treatment response. Mid-treatment monitoring of this ratio could help clinicians determine whether to maintain, intensify, or adjust the intervention based on early neurophysiological changes.”
Point 13. Consider including a correlation analysis between clinical improvements and neurophysiological changes.
Response 13. We have included in the manuscript under Table 4.
Point 14. Improve figure labeling and presentation for clarity and Correct typographical and grammatical errors throughout the manuscript.
Response 14. The manuscript was revised and we checked and corrected typographical and grammatical errors.
We would like to thank you for your effort and valuable feedback. We proofread and adjust the manuscript to comply with formatting guidelines, including citations, references, headings, tables, and figures.
Round 2
Reviewer 1 Report
Comments and Suggestions for Authors
The authors have addressed the concerns raised in my previous queries, and the revisions have enhanced the overall quality of the manuscript. My final comment is to improve the quality of the figures. The fonts in all the figures are too small, which makes the details very difficult to read.
Author Response
Dear Reviewer,
Thank you for your positive feedback. We appreciate your final observation regarding the figures. In response, we have revised all figures to increase font size and enhance overall clarity. The updated figures have been incorporated into the revised manuscript.
Thanks again for your effort and work in revising our manuscript.
Kind regards
